# Magnetic and MRI Contrast Properties of HumAfFt-SPIONs: Investigating Superparamagnetic Behavior and Enhanced T_2_-Weighted Imaging Performance

**DOI:** 10.3390/ijms26083505

**Published:** 2025-04-09

**Authors:** Luisa Affatigato, Mariano Licciardi, Maria Cristina D’Oca, Luca Cicero, Alessandra Bonamore, Alessio Incocciati, Alberto Macone, Christian Dirk Buch, Stergios Piligkos, Alberto Boffi, Valeria Militello

**Affiliations:** 1Department of Physics and Chemistry–Emilio Segrè, University of Palermo, 90128 Palermo, Italy; luisa.affatigato@unipa.it (L.A.); mariacristina.doca@unipa.it (M.C.D.); valeria.militello@unipa.it (V.M.); 2Department of Biological, Chemical and Pharmaceutical Sciences and Technologies (STEBICEF), University of Palermo, 90123 Palermo, Italy; 3Istituto Zooprofilattico Sperimentale della Sicilia–A. Mirri, 90100 Palermo, Italy; lucacicero031182@gmail.com; 4Department of Biochemical Sciences–A. Rossi Fanelli, Sapienza University, 00185 Rome, Italy; alessandra.bonamore@uniroma1.it (A.B.); alessio.incocciati@uniroma1.it (A.I.); alberto.macone@uniroma1.it (A.M.); alberto.boffi@uniroma1.it (A.B.); 5Department of Chemistry, University of Copenhagen, DK-2100 Copenhagen, Denmark; christian.buch@chem.ku.dk (C.D.B.); piligkos@chem.ku.dk (S.P.)

**Keywords:** ferritin-coated SPIONs, superparamagnetic properties, MRI contrast agent

## Abstract

The present study introduces a novel theranostic nanoparticle platform that integrates superparamagnetic iron oxide nanoparticles (SPIONs) with a ferritin-based protein nanocage derived from the archaeobacterium *Archaeoglobus fulgidus*. By exploiting the unique salt-triggered dissociation and reassociation mechanism of the nanocage, SPIONs were successfully encapsulated within the protein’s central cavity. The construct thus obtained was characterized by transmission electron microscopy and circular dichroism spectroscopy. The ferritin-coated SPIONs exhibited remarkable superparamagnetic behavior and robust magnetic properties. Characterization using electron paramagnetic resonance and thermal magnetization analysis confirmed the stability of the nanoparticles and their suitability for magnetic hyperthermia applications. Furthermore, T_2_-weighted magnetic resonance imaging (MRI) demonstrated enhanced contrast, with ferritin-coated SPIONs generating distinct dark-spot imaging, highlighting their efficacy as a contrast agent for advanced biomedical applications.

## 1. Introduction

Nanomaterials and nanodevices have revolutionized biomedical applications, offering innovative solutions for diagnostics and therapeutics [1]. Among them, SPIONs have gained significant attention for their extensive use in various in vivo applications [2,3,4,5]. These nanoparticles feature solid cores composed of iron oxides—either magnetite (Fe_3_O_4_) or maghemite (Fe_2_O_3_)—encapsulated within biocompatible polymer coatings [6]. Their remarkable physical, chemical, thermal, and mechanical properties position SPIONs as highly promising theranostic agents. They have been widely investigated for their role in cancer MRI and their potential synergy with chemotherapy and magnetotherapy approaches [7,8]. They can be directed to tumor sites using external static magnetic fields, facilitating targeted accumulation within tumor masses.

In medical imaging, SPIONs function as contrast agents for MRI [9,10,11]. This technique has many advantages over other clinical methods, such as enhanced soft tissue contrast and high spatial resolution, which improve the visualization of anatomical details, and above all, it does not involve the use of ionizing radiation, which has some important side effects. In order to enhance the quality of MRI scans, it is essential to use the right contrast agents. Traditional contrast agents, such as gadolinium-based compounds, improve T_1_-weighted images by shortening T_1_ relaxation time, leading to enhanced brightness in imaging regions [12]. Unfortunately, gadolinium is toxic to humans and has been shown to accumulate in the brain, independent of renal function, as well as in other tissues (bone, liver, spleen, skin, and kidneys). Thus, limitations such as low sensitivity and potential toxicity have prompted the search for alternative materials. SPIONs under standard MRI field strengths reach magnetic saturation, inducing significant local dipolar field perturbations. This results in a reduction in proton T_2_ relaxation times, thereby enhancing contrast in T_2_-weighted imaging [13]. Additionally, SPIONs exhibit superior magnetic susceptibility compared to other paramagnetic materials, particularly when their diameter is below 20 nm, as they maintain a single magnetic domain. Conversely, SPIONs larger than 20 nm contain multiple magnetic domains with opposing electron spins, which reduces their overall magnetic response [14]. SPIONs represent an alternative contrast agent combining enhanced contrast properties and nanotechnology. They have attracted considerable attention for their high biocompatibility, small particle size, large surface area, and intrinsic magnetic properties.

Beyond imaging, SPIONs hold significant promise for therapeutic applications such as magnetic hyperthermia, where exposure to an alternating magnetic field induces localized heat generation [15]. This feature has been demonstrated to be highly effective in eradicating cancer cells, as these cells are unable to withstand temperatures between 42 and 49 °C—a range that healthy cells can tolerate [16]. The application of an alternating magnetic field to magnetic particles results in the generation of heat due to magnetic hysteresis loss. The quantity of the generated heat is contingent upon the magnetic properties of the material and the field parameters. Other biomedical applications include tissue repair, immunoassays, drug delivery, and cell separation [17]. In all these applications, the properties of SPIONs exhibit a strong dependence on their size and shape [18]. Usually, biomedical applications require inorganic cores smaller than 100 nm.

To enhance their colloidal stability, solubility in physiological media, and biocompatibility, SPIONs are typically coated with suitable materials. Particle coating can be achieved through two distinct methods: (1) adsorption, in which small organic molecules are attached to the surface of nanoparticles using anchoring groups like silanes, carboxylates, or organophosphorus, which have high affinity with the metal oxide surface [19]; or (2) encapsulation, where SPIONs are enclosed within a self-assembled amphiphilic structure, resulting in the formation of magnetic nanoassemblies [20,21] that display excellent dispersibility and biocompatibility. Considerable effort has been dedicated to the surface functionalization of SPIONs using different coating materials. Selecting the correct coating agent is crucial, since this will affect particle cellular uptake, protein adsorption and interaction with cells and tissues, and biokinetic parameters such as biodistribution, biodegradation, metabolism, and elimination [22]. Several polymer-coated SPION-based agents, such as Feridex, Resovist, Ferumoxtran-10, and Clariscan, have been clinically approved for general imaging applications [5]. However, a major drawback of these agents is their lack of tumor-specific targeting, which limits their ability to distinguish cancerous tissues from normal ones. To address this limitation, we employed a ferritin-based protein nanocage for coating SPIONs. Ferritins are a family of proteins, found in all domains of life, which are composed of 24 subunits to form a cage architecture of 12 nm diameter with an interior cavity of 8 nm diameter [23]. Mammalian ferritins consist of a mixture of two different types of subunit known as H (heavy) and L (light) chains. The H-chain is important for Fe(II) oxidation, because it possesses a ferroxidase center which can catalyze the oxidation of Fe(II) to Fe(III) [24], while the L-chain’s lack of this center assists in iron core formation [25]. In addition to their distinctive architectural features, ferritins exhibit unique physical and chemical properties [26,27,28]. In contrast to the sensitivity of most proteins to temperature and pH outside of the physiological range, ferritin demonstrates a remarkable resistance to high temperatures of up to 80 °C for 10 min and is also able to retain its structural integrity when subjected to a variety of denaturants, including urea [29,30]. These unique properties of ferritin are attributed to a large number of salt bridges and hydrogen bonds formed between subunits. Human ferritin H-chain homopolymer is a biologically derived nanovector widely used in biomedicine due to its low toxicity and remarkable ability to selectively target tumor cells [31,32]. This selectivity is primarily driven by the interaction between an unstructured region on ferritin’s surface (the BC loop) and CD71 receptors, which exhibit nanomolar affinity [33]. Although CD71 is expressed in various cell types, it is significantly overexpressed on the surface of iron-avid cancer cells [34,35,36], making it a crucial target for ferritin-based drug delivery systems. The elevated expression of CD71 in malignant cells enhances nanoparticle binding efficiency, further supporting its role in tumor-specific therapy [37]. In this study, our focus centered on a chimeric nanoparticle derived from the ferritin of the archaeobacterium *Archaeoglobus fulgidus.* It has a non-canonical 24-mer cage with a peculiar tetrahedral geometry and four 45 Å triangular pores on the protein shell. An intriguing feature of this ferritin is its unique ability to dissociate and reassociate under mild conditions, accomplished by simply modulating the concentration of divalent cations in the medium [38,39]; unlike human ferritin, which requires instead extreme pH conditions for dissociation [40]. This salt-triggered dissociation–reassociation process is controlled solely by adjusting the magnesium ion concentration in the medium. At Mg^2+^ concentrations below 10 mM, the protein cage temporarily opens; while at 50 mM Mg^2+^, it fully reassembles. This reversible behavior enables the encapsulation of SPIONs within the cavity while preserving the structural integrity of the protein. Since bacterial ferritin does not inherently recognize the human CD71 receptor, we engineered the *Archaeoglobus fulgidus* ferritin by incorporating the human ferritin BC loop sequence, enabling tumor-specific targeting [41]. To date, the new chimeric ferritin (HumAfFt) has demonstrated successful applications in encapsulating and delivering drugs, therapeutic proteins, and small nucleic acids to tumor cells [42,43,44].

## 2. Results and Discussion

In this study, HumAfFt was used to coat SPIONs, creating HumAfFt-SPIONs—a multifunctional nanoparticle that integrates the targeting ability of HumAfFt with the theranostic properties of SPIONs. Our findings confirm that HumAfFt-SPIONs preferentially accumulate in cancer cells rather than normal cells [45]. The complex was characterized for size, morphology, and magnetic properties using electron paramagnetic resonance (EPR) and thermal magnetization analysis, demonstrating its potential for magnetic hyperthermia. Importantly, this study provides the first in-depth investigation of the superparamagnetic behavior of HumAfFt-SPIONs and their enhanced T_2_-weighted imaging performance, establishing their potential as next-generation MRI contrast agents. Additionally, the first MRI images using HumAfFt-SPIONs as a contrast agent have been generated and they highlight their potential for advanced cancer diagnostics and therapy.

### 2.1. Preparation of HumaAfFt-SPIONs

The humanized ferritin from *A. fulgidus* (HumAfFt) was overexpressed in *E. coli* and purified with high yields (Appendix A). High-Performance Size Exclusion Chromatography (HP-SEC) analysis confirmed that the purified ferritin correctly assembled into a 24-mer structure (Appendix A) and retained its characteristic dissociation/association behavior mediated by divalent cations. Its negatively charged interior and suitable size make this nanoparticle ideal for hosting superparamagnetic iron oxide nanoparticles (SPIONs) with an average size of 10 nm. To enable encapsulation, the HumAfFt was dissociated by removing magnesium chloride. The disassembled HumAfFt was incubated with SPIONs at a 1:1 concentration ratio, followed by reassembly via the addition of MgCl_2_ to a final concentration of 50 mM (Figure 1).

To purify the HumAfFt-SPIONs complex and remove unassembled ferritin, an external magnet was applied to the vial. The stable HumAfFt-SPIONs complex was rapidly collected near the magnet. Upon removal of the magnetic field, the nanocomplex dispersed, demonstrating the superparamagnetic behavior of the coated nanoparticles. The UV-vis absorption spectra of the supernatant post-purification revealed no detectable protein, confirming that all ferritin molecules were successfully complexed with SPIONs.

### 2.2. Characterization of HumAfFt-SPIONs

Circular dichroism (CD) spectroscopy was performed to characterize the secondary structure of HumAfFt both in the absence and presence of SPIONs. The spectrum of the native protein displayed the hallmark features of a well-defined, predominantly α-helical protein structure, with two distinct negative peaks at 208 nm and 222 nm (Figure 2A) [46,47,48,49,50]. These results align well with structural data obtained from the Protein Data Bank (5LS9), which indicate that HumAfFt is composed of 67.8% α-helix, 0% β-sheet, 5.7% β-turns, and 26.5% random coil. Upon encapsulation of SPIONs, the CD spectra exhibited a decrease in overall intensity as the SPION concentration increased. Since SPIONs themselves do not possess intrinsic CD signals, the observed spectral changes can be attributed to the interaction between HumAfFt and SPIONs. Notably, the general spectral features of HumAfFt remain characteristic of an α-helical-dominant structure, suggesting that the overall secondary structure of the protein is largely preserved. Given that the average diameter of SPIONs (10 nm) depends on the internal cavity size of ferritin (about 8–9 nm), the interaction between HumAfFt and SPIONs might exert mechanical stress on the protein shell, potentially causing local distortions in the tertiary structure or partial shielding of the aromatic residues. These factors could contribute to the observed decrease in CD signal intensity. To ensure the stability of the analyzed samples and to rule out precipitation during the CD measurements, complementary absorbance and fluorescence spectroscopy were performed every five minutes over a one-hour period. The absorbance at 280 nm and 600 nm remained unchanged throughout this timeframe, and no variations in the fluorescence emission spectra were observed (see an example of the absorption and emission spectra in Appendix A). These results confirm the stability of the samples and the absence of aggregation or sedimentation during the experiments.

The morphology and size of SPIONs and the resulting nanoparticle system HumAfFt-SPIONs, were characterized using transmission electron microscopy (TEM). Both SPIONs and HumAfFt-SPIONs exhibited a narrow size distribution and well-rounded morphology (Figure 2B). Specifically, the average size of SPIONs was confirmed to be 9.5 nm, which is suitable for encapsulation within the ferritin cavity. The size of HumAfFt-SPIONs was determined to be approximately 13.5 nm, consistent with values reported in the literature for HumAfFt. This indicates that SPION encapsulation preserves the expected quaternary assembly of the ferritin structure. Additionally, different facets of the nanoparticles can be distinguished by variations in color saturation, potentially corresponding to distinct planes of the crystal structure [9,51].

### 2.3. Magnetic Properties of HumAfFt-SPIONs

The magnetic properties of HumAfFt-SPIONs and uncoated SPIONs were investigated using electron paramagnetic resonance (EPR) spectroscopy. Figure 3 shows the room temperature EPR spectra of both SPIONs and freeze-dried HumAfFt-SPION samples, normalized to the mass of SPIONs for comparison.

At room temperature, a very broad and strong single asymmetric microwave resonance signal is observed in EPR spectra at a field of around 2920 G with linewidths of ≅780 G and *g*-values of 2.4 for SPIONs. The resonance field and the linewidth of the EPR signal depend on the coating material [52]. Both the resonance field and the linewidth increase due to coating. For a resonance field of around 3400 G there are linewidths of ≅1080 G and *g*-values of 2.1 for HumAfFt-SPIONs. The total effective magnetic moment of the SPIONs decreases due to coating, which is due to a non-collinear spin structure originating from pinning of the surface spins and coated ferritin at the interface of nanoparticles.

Furthermore, to investigate the effect of encapsulation on the magnetic properties of the SPIONs, zero-field-cooled (ZFC) and field-cooled (FC) magnetization measurements were performed both on dispersion in water and on lyophilized samples of SPIONs and HumAfFt-SPIONs [53]. In the whole temperature interval investigated, the FC and ZFC magnetization curves increased upon lowering the temperature. In the low-temperature part of the magnetization, the ZFC and FC curves of each system do not overlap, indicating some magnetic blocking of the magnetic particles. In both systems, the ZFC and FC magnetization curves start to overlap around 70 K, indicating the transition to a super-paramagnetic regime above this temperature. Upon examination of the ZFC and FC magnetization curves in Figure 4A, it was discovered that the magnetization curves of the HumAfFt-SPIONs demonstrated greater magnetization values at all evaluated temperatures. We attribute this difference to the samples not being completely dry. In order to better compare the behavior of SPIONs and that of HumAfFt-SPIONs, we chose to normalize the data (Figure 4B). The normalized magnetization curves are fully superimposable, showing that the magnetic properties of the SPIONs remain unchanged upon encapsulation in HumAfFt. These properties were also confirmed when the samples were in water (Figure 4C), suggesting that magnetic properties of HumAfFt-SPIONs are not affected by physical environment and interactions.

These findings suggest that the HumAfFt coating alters the local magnetic environment of SPIONs, potentially enhancing their stability and suitability for biomedical applications like MRI.

### 2.4. MRI Contrast Performance of HumAfFt-SPIONs

The ability of HumAfFt-SPIONs to act as contrast for MRI was evaluated by recording T_2_-weighted MRI images of nanoparticle dispersion [10,54,55]. The T_2_-weighted magnetic resonance images (MRI) shown in Figure 5 illustrate the behavior of HumAfFt and HumAfFt-SPIONs in aqueous dispersion. The image on the left, corresponding to HumAfFt alone, displays a relatively uniform signal intensity across the sample, with no significant dark regions, indicating the absence of materials capable of influencing the T_2_ relaxation process. In contrast, the image on the right, representing HumAfFt-SPIONs, shows prominent dark spots in specific zones (highlighted by arrows), which are characteristic of areas with significant nanoparticle accumulation. These dark regions are attributed to the superparamagnetic properties of SPIONs, which generate local magnetic field inhomogeneities, leading to enhanced T_2_ signal attenuation. The presence of these dark spots confirms the successful integration of SPIONs into the HumAfFt complex and their effective dispersion within the sample. This result highlights the potential utility of HumAfFt-SPIONs as a T_2_ contrast agent in magnetic resonance imaging applications, with the dark regions serving as markers for nanoparticle localization.

## 3. Materials and Methods

### 3.1. Expression and Purification of HumAfFt

The synthetic gene encoding for humanized *Archaeoglobus fulgidus* ferritin (HumAfFt), optimized for the expression in *Escherichia coli* cells, was synthesized by Gene Art (ThermoFisher, Dreieich, Germany) and subcloned into a pET22b vector (Novagen, Merck Milano, Italy) between the NdeI and XhoI restriction sites. The recombinant plasmid was transformed into *Escherichia coli* BL21(DE3) for protein expression upon induction with 1 mM IPTG (isopropyl-β-D-1-thiogalactopyranoside) at OD_600_ = 0.6 for 16 h. After 16 h, the bacterial cells were harvested by centrifugation and treated as previously described [56], with slight modification. Briefly, the bacterial paste was resuspended in 20 mM HEPES buffer containing 50 mM MgCl_2_ in the presence of a Protease Inhibitor Cocktail (Basel, Switzerland). After sonication, the soluble fraction was subjected to two (NH_4_)_2_SO_4_ precipitations (50% and 70%). The pellet from the 70% (NH_4_)_2_SO_4_ was recovered by centrifugation and extensively dialyzed versus 10 mM sodium phosphate buffer of pH 7.2 containing 20 mM MgCl_2_. After dialysis, the sample was heated at 78 °C and centrifuged to remove the denatured protein contaminants. The sample was then digested with deoxyribonuclease I from a bovine pancreas (Merck, Darmstadt, Germany) and DNA removal was achieved in a single step by means of crossflow ultrafiltration using a single Vivaflow 200 module (Sartorius) with a cutoff of 100 kDa, coupled to a Masterflex L/S pump system. Crossflow ultrafiltration was also used in diafiltration mode to exchange the buffer with 20 mM HEPES of pH 7.4 containing 50 mM MgCl_2_. Protein purity was monitored by SDS-PAGE (Biorad, Hercules, CA, USA). DNA removal was followed by measuring a 260/280 nm ratio using a Jasco V-650 spectrophotometer (JASCO, Hessen, Germany).

As the final purification step, the protein was loaded onto a HiLoad 26/600 Superdex 200 pg column previously equilibrated in the same buffer using an ÄKTA-Prime system (GE Healthcare, Chicago, IL, USA) following the absorbance at 280 nm. Purified protein was concentrated to 1 mg/mL using AmiconUltra-15 centrifugal filter devices (100 kDa cut-off). Protein concentration was determined using an extinction coefficient of 32,430 M^−1^cm^−1^ at 280 nm. The protein purity was assessed by SDS-PAGE and the correct assembly by analytical size-exclusion chromatography using an Agilent AdvanceBio SEC 300 Å (Agilent, Santa Clara, CA, USA), 7.8 × 150 mm, 2.7 µm, LC column in isocratic mode at a flow rate of 1 mL/min with a mobile phase composition of 20 mM HEPES and 50 mM MgCl_2_ of pH 7.4. (UV detection at 220 and 280 nm).

### 3.2. Synthesis of SPIONs Coated with HumAfFt

Superparamagnetic iron oxide nanoparticles (SPIONs) with an average particle size of 10 nm, dispersed in water (5 mg/mL, approximately 3 µM SPIONs), were obtained from Sigma Aldrich (Milan, Italy). To disassemble HumAfFt (3 µM), MgCl_2_ was removed via dialysis against demineralized Milli-Q water using a Spectral/Por Dialysis Membrane (Standard RC Tubing, MWCO: 3.5 kD, Merck Milano, Italy). Subsequently, HumAfFt was reassembled with SPIONs in a 1:1 molar ratio by reintroducing MgCl_2_ to a final concentration of 50 mM. The resulting HumAfFt-SPION complexes were collected and purified using an external magnetic field. The presence of residual HumAfFt in the solution was assessed by UV absorption spectroscopy. The dispersion was freeze-dried for further analysis, confirming the absence of free ferritin in solution.

### 3.3. Circular Dichroism

Circular dichroism (CD) spectroscopy was performed on both HumAfFt and HumAfFt-SPIONs using a Chirascan CD spectrophotometer (Applied Photophysics, Surrey, UK) with an LAAPD detector and Chirascan Spectrometer Control Panel software version 4.4 (Applied Photophysics, Leatherhead, UK). Far-UV CD analysis was performed from 190 to 260 nm with a 0.5 nm step size. All measurements were performed at 25 °C using a 1 mm precision cuvette (Hellma GmbH, Müllheim, Germany), and each sample was scanned 3 times and averaged. No smoothing was used. All sample solvents were also scanned under identical conditions and subtracted from the sample spectra. Raw data were analyzed using BeStSel-Protein Circular Dichroism Spectra Analysis (https://bestsel.elte.hu/index.php, accessed on 30 June 2023).

### 3.4. Absorption and Emission Measurements

HumAfFt and HumAfFt-SPION samples (500 μL) were scanned in a quartz cuvette with 3 × 3 mm light path (Hellma Analytics, Müllheim, Germany) while measuring the absorbance at 225–600 nm and fluorescence at 281–800 nm with excitation at 280 nm. The measurements were taken every 60 s for one hour to analyze the stability of the samples. The spectra were obtained using a Labbot instrument (ProbationLabs, Lund, Sweden).

### 3.5. Trasmission Electron Microscopy

Aliquots (3 μL) of SPIONs (5 mg/mL) and HumAfFt-SPION samples (0.03 mg/mL) were deposited on a hydrophilized carbon grid (pure C, 200 mesh Cu, Pelco, Fresno, CA, USA). Images were taken using a CM 100 TEM (Philips, the Netherlands), operated at 80 kV, connected to an Olympus Veleta camera (Göttingen, Germany) (resolution—2048 × 2048 pixels (2 K × 2 K)) and the iTEM software package (Olympus Soft Imaging Solutions GMBH, Münster, Germany).

High-magnification images were analyzed by the freeware software s (version 1.29, NIH, Bethesda, Rockville, MD, USA). Measurements of the areas with more than 100 particles were taken for each sample of SPIONs and HumAfFt-SPIONs. The average diameter distribution was further calculated by OriginLab.

### 3.6. Electron Paramagnetic Resonance (EPR)

Electron paramagnetic resonance (EPR) spectra of SPIONs and HumAfFt-SPIONs transferred into sealed tubes of 5 mm internal diameter were recorded using an X band (microwave frequency ≅ 9.8 GHz) from Bruker ELEXSYS E 500 (Karlsruhe, Germany), with an ER 4122SHQ type cavity resonator. EPR spectra of the samples were displayed in the form of the first derivative of the absorption peak plotted against the magnetic field. All measurements were made at room temperature (291 K). Spectrometer operating conditions are 9.85 GHz, 3300 field set, 100 Hz field modulation, 4 G peak-to-peak modulation amplitude and 2.4 mW microwave power to determine signal linewidths and *g*-values.

### 3.7. Magnetic Characterization

Measurements of direct current (DC) magnetization as a function of temperature were performed on a Quantum Design MPMS-XL superconducting quantum interference device (SQUID) (San Diego, CA, USA). For the zero-field-cooled (ZFC) measurements, the sample was cooled to 2 K without an applied magnetic field, and then the magnetization was recorded in the temperature range 2–260 K with an applied static magnetic field of 1000 Oe. For the field-cooled (FC) measurements, the sample was cooled down to 2 K in an applied static magnetic field of 1000 Oe, and then the magnetization was measured in the temperature range 2–260 K, retaining the static magnetic field.

The ZFC and FC measurements were performed on both dry samples of SPIONs and HumAfFt-SPIONs as well as 150 mg/mL concentrations of these in water. For the dry samples a small amount of n-hexadecane was used to fixate the samples to avoid orientation in the magnetic field.

### 3.8. Magnetic Resonance Imaging (MRI) at 7 T

MRI measurements of HumAfFt and HumAfFt-SPIONs in aqueous dispersion were performed with a 7 T horizontal bore PharmaScan 70/16 US scanner (Bruker, Ettlingen, Germany) using a 23 mm transmit/receive volumetric coil. A respiration-triggered gradient echo sequence (TE) = 35.0 ms and a triggered T_2_-Turbo RARE sequence were selected for T_2_ imaging.

## 4. Conclusions

This study highlights the potential magnetic and MRI contrast properties of HumAfFt-SPIONs for biomedical applications, demonstrating their superparamagnetic behavior and enhanced performance in T_2_-weighted imaging. By exploiting the unique salt-triggered assembly–disassembly mechanism of HumAfFt and its intrinsic targeting capability via the transferrin receptor, we successfully encapsulated SPIONs within the protein cage. This process preserved both the structural integrity and biological functionality of the ferritin cage, ensuring that the resulting nanoparticles maintained their biocompatibility and targeting efficiency. The synthesized HumAfFt-SPIONs exhibit remarkable physicochemical properties, including high colloidal stability, uniform size distribution, and robust magnetic responsiveness. Their enhanced contrast in T_2_-weighted MRI was confirmed through dark-spot imaging, demonstrating their ability to produce significant signal attenuation and improve tumor visualization. These attributes position HumAfFt-SPIONs as a highly promising dual-functional platform with both diagnostic and therapeutic applications. Beyond their role as an efficient MRI contrast agent for precise tumor imaging, HumAfFt-SPIONs hold significant potential for future biomedical applications, particularly in targeted cancer imaging and therapy. Their unique combination of high biocompatibility, tumor-targeting ability via CD71 overexpression, and strong superparamagnetic behavior enables enhanced T_2_-weighted MRI contrast and potential magnetic hyperthermia applications. Compared to conventional MRI contrast agents, such as gadolinium-based compounds [57,58], HumAfFt-SPION complexes potentially offer a safer alternative with reduced toxicity risks. Moreover, their ability to encapsulate therapeutic agents within the ferritin cavity provides a multifunctional platform for theranostics. However, the clinical translation of ferritin-based nanocarriers remains challenging due to factors such biodistribution, pharmacokinetics, clearance, and regulatory hurdles [59,60]. Recent studies have explored similar protein-based nanoparticles for imaging and drug delivery [61,62,63], demonstrating promising preclinical results but highlighting the need for further pharmacokinetic and safety evaluations before clinical application. Addressing these challenges through in vivo studies will be crucial to advancing HumAfFt-SPIONs toward clinical use.

## Figures and Tables

**Figure 1 ijms-26-03505-f001:**
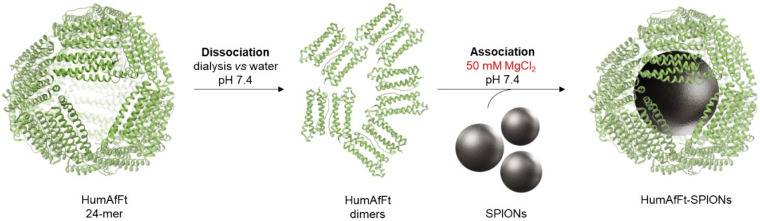
Schematic representation of the encapsulation process of the SPIONs. The disassembled HumAfFt was incubated in the presence of the SPIONs and the association/encapsulation step was triggered by adding MgCl_2_ (50 mM final concentration).

**Figure 2 ijms-26-03505-f002:**
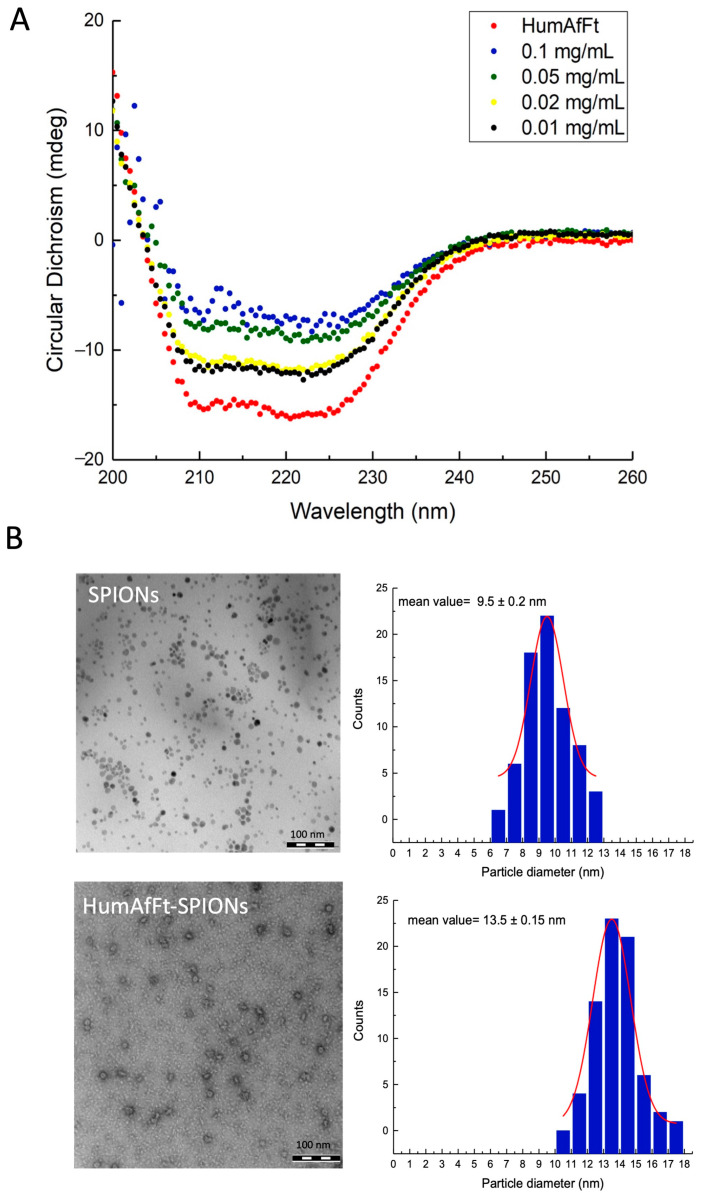
Characterization of HumAfFt-SPIONs. (**A**) Circular dichroism curves for HumAfFt (0.1 mg/mL) and HumAfFt-SPIONs at different concentrations of SPIONs (0.1 mg/mL; 0.05 mg/mL; 0.02 mg/mL; 0.01 mg/mL). (**B**) TEM images of SPIONs and HumAfFt-SPIONs with size distribution.

**Figure 3 ijms-26-03505-f003:**
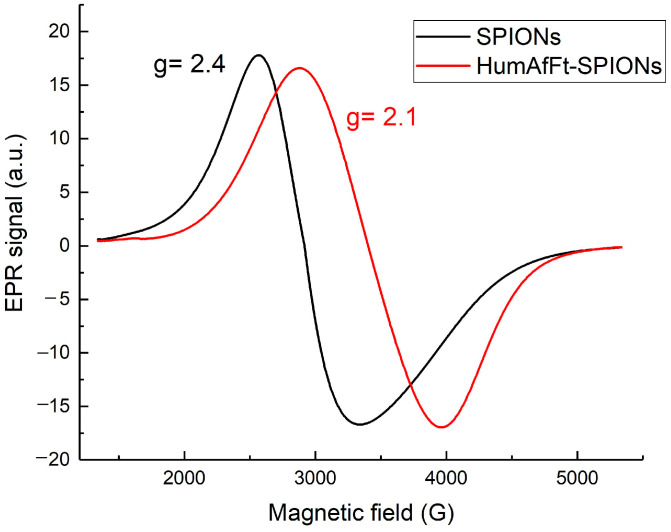
Room temperature EPR signals for SPIONs and HumAfFt-SPION samples.

**Figure 4 ijms-26-03505-f004:**
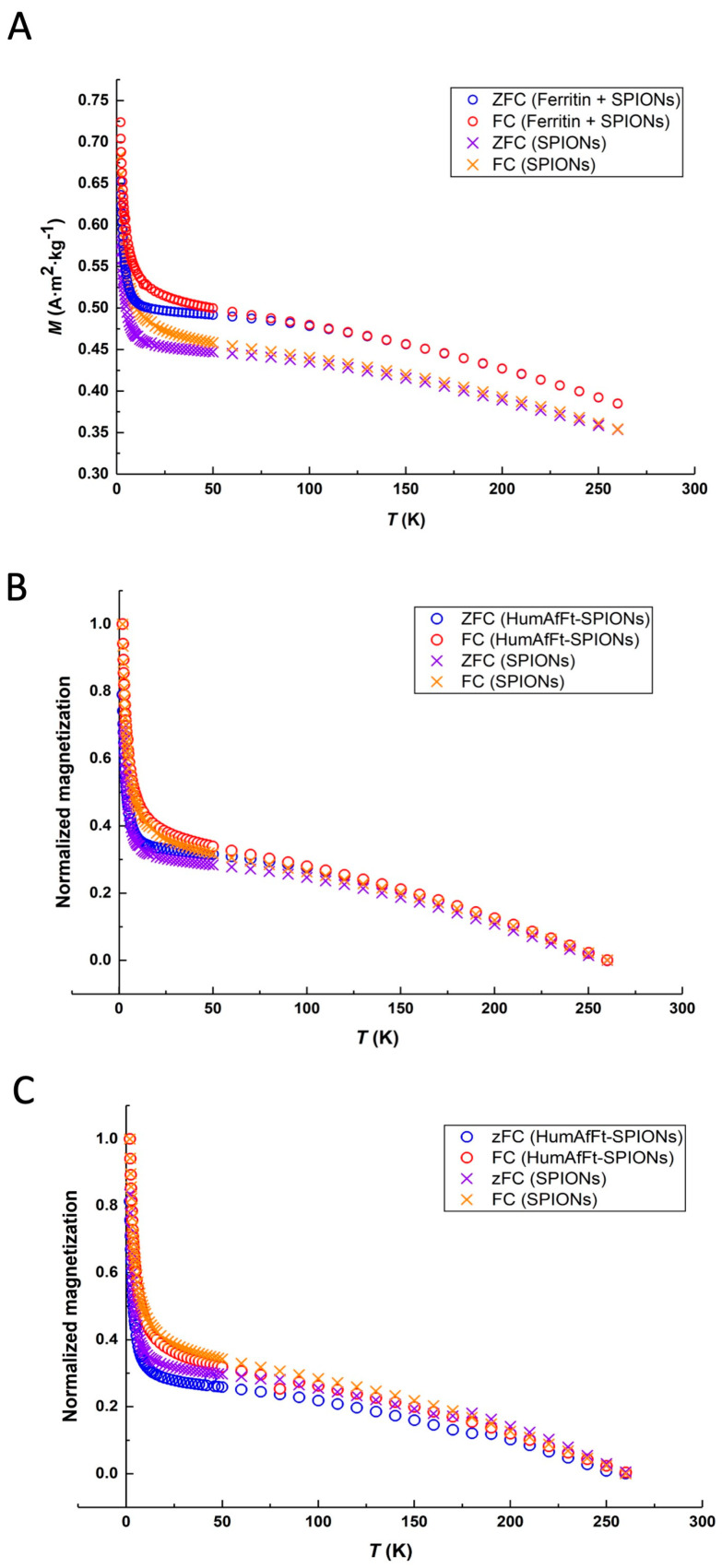
(**A**) ZFC and FC magnetization curves (magnetic field of 1000 G) for lyophilized SPIONs and HumAfFt-SPIONs. (**B**) Normalized ZFC and FC magnetization curves (static magnetic field of 1000 Oe) for lyophilized SPIONs and HumAfFt-SPIONs. (**C**) Normalized ZFC and FC magnetization curves for SPIONs and HumAfFt-SPIONs in water.

**Figure 5 ijms-26-03505-f005:**
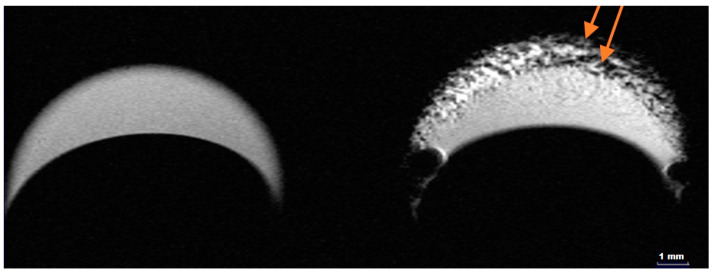
T_2_-weighted images of HumAfFt (**left**) and HumAfFt-SPIONs (**right**) in aqueous dispersion. The arrows highlight representative regions, with dark spots indicating areas of nanoparticle accumulation.

## Data Availability

The data that support the findings of this study are available from the corresponding author upon reasonable request.

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
