# Peer review of "Magnetic and MRI Contrast Properties of HumAfFt-SPIONs: Investigating Superparamagnetic Behavior and Enhanced T2-Weighted Imaging Performance"

_ijms, 2025, doi:10.3390/ijms26083505_

Round 1

Reviewer 1 Report

Comments and Suggestions for Authors

Review Report: “Magnetic and MRI Contrast Properties of HumAfFt-SPIONs: Investigating Superparamagnetic Behavior and Enhanced T2-Weighted Imaging Performance”

The manuscript “Magnetic and MRI Contrast Properties of HumAfFt-SPIONs: Investigating Superparamagnetic Behavior and Enhanced T2-Weighted Imaging Performance” by Luisa Affatigato et al. reported the new theranostic nanoparticle platform to target the SPINs, the so called integrating Superparamagnetic Iron Oxide Nanoparticles, with  a humanized ferritin-based protein nanocage derived from Archaeoglobus fulgidus (HumAfFt). Overall, this manuscript is well structured and easy to follow. The experimental demonstration is solid and sound. They demonstrated the HumAfFt-SPIONs is a promising approach to give an alternative to MRI contrast agents. Those strong evidence suggest this new approach could potentially have great vision and applications in the future, I suggested this manuscript to be published. Here I also list some detailed suggestions:

  • The experimental result demonstrated a great characterization on in vitro condition, would the new approach has similar performance in the in vivo experiments? Saying, compared with MRI contrast method?
  • How long can the SPIONs last without aging? The magnetic properties would be stabilized under what conditions?
  • The discussion of the experiments regarding CD71 cancer is vague.
  • Figure 2’s label are too small to read. What about temperature dependent data for the CD measurements? Why different wavelength would give the different sign of the CD signal? CD should probe the time-reversal symmetry breaking, but how can wavelength change the sign of the signal?
  • Could the author add more info in the Fig. 4’s caption, like the FC curves, under how large field and the orientation of the field?
  • 5 should be remade, add scale bar that could be easily read.

Overall, this is a solid research with sound potential for future applications.

Author Response

REVIEWER 1

The manuscript “Magnetic and MRI Contrast Properties of HumAfFt-SPIONs: Investigating Superparamagnetic Behavior and Enhanced T2-Weighted Imaging Performance” by Luisa Affatigato et al. reported the new theranostic nanoparticle platform to target the SPIONs, the so called integrating Superparamagnetic Iron Oxide Nanoparticles, with a humanized ferritin-based protein nanocage derived from Archaeoglobus fulgidus (HumAfFt). Overall, this manuscript is well structured and easy to follow. The experimental demonstration is solid and sound. They demonstrated the HumAfFt-SPIONs is a promising approach to give an alternative to MRI contrast agents. Those strong evidence suggest this new approach could potentially have great vision and applications in the future, I suggested this manuscript to be published. Here I also list some detailed suggestions:

  • The experimental result demonstrated a great characterization on in vitro condition, would the new approach has similar performance in the in vivo experiments? Saying, compared with MRI contrast method?

It is authors’ opinion that the new system HumAfFt-SPIONs will have great potentiality in vivo, based on the promising results we have obtained in previous studies with cells and in general on the use of SPIONs as MRI contrast agent (already approved by the FDA). Anyway, we are planning to perform in vivo studies, that are crucial to advancing HumAfFt-SPIONs toward clinical use. 

  • How long can the SPIONs last without aging? The magnetic properties would be stabilized under what conditions?

SPIONs can remain stable for months to years without significant aging, depending on several factors, including their synthesis method, surface coating and storage conditions. SPIONs without a protective coating tend to oxidize over time, leading to a loss of magnetization and increased aggregation. Functionalizing with stabilizers prevents aggregation and maintains the magnetic response. For this reason, we used HumAfFt to coat SPIONs, enabling the new nanocomplex stability and tumor-specific targeting. Actually, our sample in house are still stable after more than one year from their preparation.

  • The discussion of the experiments regarding CD71 cancer is vague.

According to reviewer’s comment, the text has been revised to better clarify the role of CD71 in cancer (line 101 to 109 of revised manuscript).

  • Figure 2’s label are too small to read. What about temperature dependent data for the CD measurements? Why different wavelength would give the different sign of the CD signal? CD should probe the time-reversal symmetry breaking, but how can wavelength change the sign of the signal?

The authors thank the reviewer for the suggestion. Accordingly, figure 2 was enlarged.

Regarding temperature-related CD measurements, authors well know that CD spectra are highly temperature-dependent, because temperature may influence molecular conformation, electronic transitions, and intermolecular interactions. For example, in biological molecules, increasing temperature can disrupt secondary structures (e.g., α-helices and β-sheets), leading to changes in CD signal intensity and shape. In our case, we carried out measurements only at room temperature, which is the temperature used for their preparation. 

Moreover, it is authors’ opinion that obtained CD spectra are well coherent with HumAfFt-SPIONs complex, as described in the paragraph 2.2.

Protein CD spectra include two types of electronic transitions arising from peptide absorptions in the far UV (A.J. Miles, R.W. Janes, B.A. Wallace, Tools and methods for circular dichroism spectroscopy of proteins: A tutorial review, Chem Soc Rev 50 (2021) 8400–8413. https://doi.org/10.1039/d0cs00558d). These include an n-p* transition that gives rise to a signal at ~222 nm, and parallel and perpendicular p-p* transitions that give rise to signals at lower wavelengths in the range from ~190–210 nm. In addition, an intra-amide charge transfer transition arising from through-space interactions occurs at around 180 nm, which is generally only accessible using synchrotron radiation as a light source. The different types of secondary structures that are adopted by polypeptide chains (for example, helix or beta strand) depend on the dihedral angles between adjacent residues in the polypeptide chain; the magnitudes and wavelengths of the CD signals that arise from the electronic transitions depend upon these angles and therefore upon the secondary structure of the protein. The overall shape and magnitude of a protein CD spectrum in the far ultraviolet (UV) wavelength region reflects a linear combination of its constituent secondary structural elements.

  • Could the author add more info in the Fig. 4’s caption, like the FC curves, under how large field and the orientation of the field?

According to reviewer’s suggestion, the above informations were added in the caption of Figure 4.

  • 5 should be remade, add scale bar that could be easily read.

According to reviewer’s suggestion, figure 5 was remade and the scale bar was added.

Overall, this is a solid research with sound potential for future applications.

Reviewer 2 Report

Comments and Suggestions for Authors

This paper, entitled “Magnetic and MRI Contrast Properties of HumAfFt-SPIONs: Investigating Superparamagnetic Behavior and Enhanced T2-Weighted Imaging Performance” introduces a novel theranostic nanoparticle platform that integrates SPIONs with a ferritin-based protein nanocage derived from the archaeobacterium Archaeoglobus fulgidus. Comprehensive characterization techniques, including transmission electron microscopy and circular dichroism spectroscopy, confirmed the structural integrity of the construct. Magnetic analyses demonstrated the stability and strong superparamagnetic behavior of the NPs, making them suitable for magnetic hyperthermia.

The authors used appropriate methodology and experimental design to test the study hypothesis.

The paper is well-organized, easily readable, and presented in a well-structured manner. The figures are appropriate and easy to understand. The conclusions are consistent with the arguments presented by the authors of this paper.

The bibliography used is well-documented, and the references are appropriate for the paper.

Therefore, I recommend that the authors address the following aspects to enhance the quality of their study before publication.

  1. Some sentences are too long and can be reworded for better readability. Expression can be made clearer in some places by removing repetitions.

  1. According to the journal formatting instructions: “References should be numbered in order of appearance and indicated by a numeral or numerals in square brackets—e.g., [1] or [2,3], or [4–6].” Please proofread the entire manuscript accordingly.

  1. The authors should emphasize more clearly the novelty of their work.

  1. Please specify in the text from the first appearance (abstract), what each abbreviated notation represents.

  1. I think that the S1A/S1B figures specified intext at Line131/Line133 should appear physically in the manuscript, not as additional information.

  1. I think that the data specified in the paragraph between lines 169 and 171 should be presented in the manuscript. These results confirm the stability of the samples.

  1. The figures in Figure 2 should be enlarged.

  1. The dispersion of the sample (in water or lyophilized) should be clearly specified in the caption of Figure 4. Figure 2S (Supplementary Material) should be introduced as the third panel in Figure 4. Alternatively, one can leave only normalized data for these two types of records (in water or lyophilized).

  1. In the Results and Discussions section, the authors should add a discussion of their results in relation to other similar results reported in the literature.

  1. In the present paper, the authors refer to the "Salt-triggered assembly-disassembly mechanism"–a brief explanation of this mechanism introduced in the paper would be useful.
  2. The statement "hold great potential for future biomedical applications" is too general. The authors should refer to similar studies or challenges in translation to clinical applications.

  1. The limitations of this study should be included in the conclusion section.

  1. The expression "exceptional magnetic and MRI contrast properties" is vague. Specific quantification or comparison with other existing contrast agents would be useful.

  1. Additionally, the similarity percentage of the work should be much lower.

  1. The authors should add a graphical abstract that highlights their work to a broader audience.

Comments on the Quality of English Language

Some sentences are too long and can be reworded for better readability. Expression can be made clearer in some places by removing repetitions.

Author Response

REVIEWR 2

Comments and Suggestions for Authors

This paper, entitled “Magnetic and MRI Contrast Properties of HumAfFt-SPIONs: Investigating Superparamagnetic Behavior and Enhanced T2-Weighted Imaging Performance” introduces a novel theranostic nanoparticle platform that integrates SPIONs with a ferritin-based protein nanocage derived from the archaeobacterium Archaeoglobus fulgidus. Comprehensive characterization techniques, including transmission electron microscopy and circular dichroism spectroscopy, confirmed the structural integrity of the construct. Magnetic analyses demonstrated the stability and strong superparamagnetic behavior of the NPs, making them suitable for magnetic hyperthermia.

The authors used appropriate methodology and experimental design to test the study hypothesis.

The paper is well-organized, easily readable, and presented in a well-structured manner. The figures are appropriate and easy to understand. The conclusions are consistent with the arguments presented by the authors of this paper.

The bibliography used is well-documented, and the references are appropriate for the paper.

Therefore, I recommend that the authors address the following aspects to enhance the quality of their study before publication.

  • Some sentences are too long and can be reworded for better readability. Expression can be made clearer in some places by removing repetitions.

The authors thank the reviewer for the valuable suggestion. Accordingly, the manuscript was improved in the revised version. In particular, repetitions were removed when possible, and the text was made more fluent in the parts highlighted in red.

  • According to the journal formatting instructions: “References should be numbered in order of appearance and indicated by a numeral or numerals in square brackets—e.g., [1] or [2,3], or [4–6].” Please proofread the entire manuscript accordingly.

References were formatted in the text of entire manuscript, as indicated by journal instructions.

  • The authors should emphasize more clearly the novelty of their work.

In order to emphasize the novelty of this work, a new clarifying sentence was added in the Results and Discussion section. Moreover, Conclusions section was expanded to better highlight the potentiality and novelty of the study.

  • Please specify in the text from the first appearance (abstract), what each abbreviated notation represents.

Accordingly, all abbreviations were explained in the text of revised manuscript.

  • I think that the S1A/S1B figures specified intext at Line131/Line133 should appear physically in the manuscript, not as additional information.

In the first draft of the manuscript, we had considered this possibility. Then, however, we decided to include the above figures as supplementary materials, because the purification is a standard procedure already developed in other publications.

  • I think that the data specified in the paragraph between lines 169 and 171 should be presented in the manuscript. These results confirm the stability of the samples.

The authors thank the reviewer for the suggestion. Accordingly, an example of the absorption and emission spectra of HumAfFt-SPIONs complex was shown in Figure S2A-B. We have decided not to include in the manuscript the data confirming the stability of our because they consist of absorption and fluorescence measurements repeated every five minutes for an hour for each sample. So, they include a large amount of graphs that could weigh down the text.

  • Figure 2 should be enlarged.

The authors thank the reviewer for the suggestion. Accordingly, figure 2 was enlarged.

  • The dispersion of the sample (in water or lyophilized) should be clearly specified in the caption of Figure 4. Figure 2S (Supplementary Material) should be introduced as the third panel in Figure 4. Alternatively, one can leave only normalized data for these two types of records (in water or lyophilized).

According to the reviewer's suggestion, the graph of normalized ZFC and FC magnetization curves for SPIONs and HumAfFt-SPIONs in water was added in main text as Figure 4C. Moreover, the physical state of the samples was specified in the caption.

  • In the Results and Discussions section, the authors should add a discussion of their results in relation to other similar results reported in the literature.

The authors agree with the reviewer that a comparison with similar study reported in the literature should better highlight the novelty of our study. Currently, no similar study, including the use of SPIONs and ferritin as superparamagnetic and contrast agent tool have been found in the literature. In the Introduction section of the manuscript, many references were cited regarding several polymer-coated SPION-based agents, such as Feridex, Resovist, Ferumoxtran-10, and Clariscan (R. Lapusan, R. Borlan, M. Focsan, Advancing MRI with magnetic nanoparticles: a comprehensive review of translational research and clinical trials, Nanoscale Adv 6 (2024) 2234–2259). The cited polymer-coated SPION systems have been clinically approved for general imaging applications; however, a major drawback of these agents is their lack of tumor-specific targeting, which limits their ability to distinguish cancerous tissues from normal ones. This finding encouraged us to propose the HumAfFt-SPIONs complex as an alternative and potentially useful colloidal system for future biomedical applications. Anyway, further text including a comparison with the similar study reported in the literature was added in the Conclusion section of the revised manuscript.

  • In the present paper, the authors refer to the "Salt-triggered assembly-disassembly mechanism"–a brief explanation of this mechanism introduced in the paper would be useful.

The Introduction section has been revised to better explain the role of salt in controlling the association–dissociation mechanism of the ferritin nanocage (lines 116-120).

  • The statement "hold great potential for future biomedical applications" is too general. The authors should refer to similar studies or challenges in translation to clinical applications.

We have provided a more detailed explanation of the potential of our study comparing it with similar studies and incorporating relevant references in the Conclusions section (lines 389-403 of the revised manuscript).

  • The limitations of this study should be included in the conclusion section.

We included them in the conclusion section (lines 397-403).

  • The expression "exceptional magnetic and MRI contrast properties" is vague. Specific quantification or comparison with other existing contrast agents would be useful.

According to the reviewer’s comment, the above sentence was changed in the Conclusion section (lines 376-377): “This study highlights potential magnetic and MRI contrast properties of HumAfFt-SPIONs, useful for biomedical application” and a comparison with other existing contrast agents was added in the conclusion section (lines 394-396) incorporating relevant references.

  • Additionally, the similarity percentage of the work should be much lower.

Accordingly, the similarities in the text were reduced when possible.

  • The authors should add a graphical abstract that highlights their work to a broader audience.

A graphical abstract has been uploaded during the submission of the revised manuscript.

Round 2

Reviewer 2 Report

Comments and Suggestions for Authors

After reviewing the authors’ responses and the modifications made to the manuscript, I am satisfied with the revisions. Therefore, I recommend the paper in this form for publication.